Impact of phenytoin and valproic acid on cytotoxicity and inflammatory mediators in human mononuclear cells: with and without lipopolysaccharide stimulation

Alesawy Aminah 1
Alotaibi Norah 1
Alalshaikh Marwa 1
Aljofi Faisal E. 1
Aldossary Nada 2
Al-Zahrani Nada 3
Omar Omar 4
Madi Marwa mimadi@iau.edu.sa 1
1 Department of Preventive Dental Sciences, College of Dentistry, Imam Abdulrahman Bin Faisal University , Dammam , Saudi Arabia
2 Department of Pathology, College of Medicine, Imam Abdulrahman Bin Faisal University , Dammam , Saudi Arabia
3 Blood Bank, Laboratory Medicine, King Fahad University Hospital , Al Khobar , Saudi Arabia
4 Department of Biomedical Dental Sciences, College of Dentistry, Imam Abdulrahman Bin Faisal University , Dammam , Saudi Arabia
Leppik Liudmila
Electronic publication date: 2025 Mar 17
Publication date: 2025
Volume: 13
Electronic Location ID: e19102
Received 2024 Oct 8; Accepted 2025 Feb 12
Copyright: ©2025 Alesawy et al.
Copyright year: 2025
Copyright holder: Alesawy et al.
License: This is an open access article distributed under the terms of the Creative Commons Attribution License, which permits unrestricted use, distribution, reproduction and adaptation in any medium and for any purpose provided that it is properly attributed. For attribution, the original author(s), title, publication source (PeerJ) and either DOI or URL of the article must be cited.
License URL: https://creativecommons.org/licenses/by/4.0/

Keywords: Phenytoin, Valproic acid, Antiepileptic drugs, Mononuclear cells, Cytotoxicity, Inflammatory mediators, IL-18, IL-6, IL-1β, IgA

Funding: The authors received no funding for this work.

==============================
Background

Valproic acid (VPA) is known for its broad-spectrum antiepileptic effects and is recommended for generalized epilepsy, in contrast to phenytoin, which has a more limited spectrum. This study investigated the cytotoxic and inflammatory responses to phenytoin and VPA in peripheral blood mononuclear cells (PBMCs), with and without bacterial lipopolysaccharide (LPS) stimulation.

Methods

PBMCs from healthy donors were divided into 12 groups: control (Ctrl), phenytoin (Phy), and four concentrations of VPA (Val-50, Val-75, Val-100, Val-200), with and without LPS. Assessments were conducted on days 1 and 3, including total, live, and dead cell counts, cell viability, and lactic acid dehydrogenase (LDH) cytotoxicity assays. Inflammatory mediators (IL-6, IL-1β) and immune markers (IL-18, IgA) were measured using enzyme-linked immunosorbent assay (ELISA) on day 3. Statistical analysis involved two-way ANOVA, Tukey’s HSD tests, and paired t-tests.

Results

All treatment groups showed significant declines in cell counts and viability from day 1 to day 3, which were exacerbated by LPS. Val-50 + LPS maintained higher cell counts compared to Ctrl + LPS and Phy + LPS. Elevated LDH levels were primarily observed in the Val-100 and Val-200 groups, with and without LPS. In the absence of LPS, the Val-75 and Val-100 groups showed notable reductions in IL-18 and IgA levels, while all VPA treatments reduced IL-6 levels compared to controls. This effect was enhanced under LPS exposure, although IL-1β reductions in the Val-75, Val-100, and Val-200 groups were reversed in the presence of LPS. Val-75 demonstrated lower cytotoxic and inflammatory responses compared to Phy and higher VPA doses, showing moderate LDH increases and reduced IL-18, IgA, IL-1β, and IL-6 levels, particularly under LPS challenge.

Conclusion

Phenytoin and VPA induced significant cytotoxic and inflammatory responses, influenced by dosage and LPS exposure. Val-75 exhibited a dose-specific immunomodulatory effect, reducing both pro-inflammatory and immune markers.

Introduction

Phenytoin is one of the most commonly used anticonvulsant medications for various conditions, including epilepsy, cerebral palsy, neuropathic pain, and bipolar disorder (Patsalos, Spencer & Berry, 2018). However, studies have shown that phenytoin treatment is frequently associated with fibrotic drug-induced gingival overgrowth (Subramani et al., 2013). In comparison to phenytoin, other anticonvulsant agents have been reported to offer better tolerability and pharmacokinetic properties (Abou-Khalil, 2019), as well as a lower incidence of gingival overgrowth (Hallmon & Rossmann, 1999), without inducing significant toxicity (Abou-Khalil, 2019).

Valproic acid (VPA) is recognized for its broad-spectrum antiepileptic effects, making it effective for nearly all types of seizures across different age groups. It remains the drug of choice for generalized epilepsy (Patsalos, Spencer & Berry, 2018). Due to its diverse therapeutic applications, VPA is also used to treat various neurological and psychiatric disorders, such as bipolar disorder and migraine (Rahman, Awosika & Nguyen, 2025). Clinically, VPA presents a lower risk of gingival enlargement. This was demonstrated in a study by Seymour, Smith & Turnbull (1985), which found a significantly higher incidence of gingival overgrowth with phenytoin compared to sodium valproate or healthy controls. These findings suggest that sodium valproate may be a safer alternative anticonvulsant with respect to gingival health.

Cytotoxicity and inflammatory responses are critical factors in evaluating the safety and efficacy of antiepileptic drugs (AEDs), as these drugs can influence immune cell behavior and tissue integrity. Both phenytoin and valproic acid (VPA) have been shown to affect cell viability, immune signaling, and inflammatory mediator levels, which can impact epilepsy treatment and overall immune function (Romoli et al., 2019; Monti, Polazzi & Contestabile, 2009; Löscher, 2002).

Specifically, AEDs like phenytoin can exacerbate neuroinflammation by increasing cytokine levels, such as IL-6, which plays a role in acute-phase responses and chronic inflammatory diseases (Vezzani & Granata, 2005). In contrast, VPA has demonstrated immunomodulatory effects, often reducing the production of pro-inflammatory cytokines (Ximenes et al., 2013).

The underlying mechanisms of gingival overgrowth associated with these medications are not fully understood. However, the complex pathophysiology of drug-induced gingival overgrowth (DIGO) involves interactions between cellular and extracellular components, influenced by factors such as age, genetic predisposition, drug pharmacokinetics, and gingival inflammation (Gawron et al., 2016). Fibroblasts, the principal cells in gingival connective tissue, play a key role in the formation and turnover of the extracellular matrix, a process regulated by bioactive molecules within the local tissue environment (Myrillas et al., 1999).

Previous studies have shown that both phenytoin and VPA affect macrophage function, inflammation, and potentially fibrosis, which may contribute to the gingival overgrowth observed clinically (Beghi & Shorvon, 2011). It has been suggested that phenytoin and VPA induce alterations in the immune system and cytokine profiles within gingival tissues, leading to dysregulation of connective tissue turnover. This dysregulation can result in the accumulation of matrix components, initiating fibrotic gingival overgrowth. Despite these similarities, differences between the two drugs stem from variations in their mechanisms of action and pharmacokinetic properties, which influence patient susceptibility to immune reactions and the likelihood of developing gingival overgrowth.

Phenytoin has been shown to decrease suppressor T cells and induce reversible IgA deficiency in patients with epilepsy. As a result, phenytoin-induced gingival overgrowth is likely attributed to increased production of IL-6 and IL-8, in conjunction with elevated basic fibroblast growth factor (FGF). This rise in inflammatory mediators promotes the recruitment and activation of inflammatory cells, facilitating interactions between cytokines and periodontal connective tissue cells (Modéer et al., 2000).

In contrast, the available data on valproate’s effects on the immune system remains inconsistent (Beghi & Shorvon, 2011). One study reported diverse effects of VPA on cytokine production, including decreased TNF-α and IL-6 production by monocytes and glial cells, along with increased levels of IL-1α, IL-1β, and IL-6 in pediatric patients, with no observed change in IL-2 production (Verrotti et al., 2001). Several studies have indicated that VPA significantly modulates the immune response and inflammation in vitro, exhibiting anti-inflammatory effects, reducing cell proliferation, and inducing apoptosis in microglial cells (Ximenes et al., 2013; Singh et al., 2021).

Among several cytokines, IL-1β and IL-6 play key regulatory roles in gingival and periodontal connective tissue turnover (Ganesh, 2016). IL-6 appears to have an autoregulatory function, influencing both pro-fibrotic and pro-inflammatory cytokines. It promotes connective tissue accumulation by stimulating the production of tissue inhibitors of metalloproteinases (TIMP) (Yamada et al., 2000). These cytokines are released by various cell types, including monocytes/macrophages, fibroblasts, and epithelial cells. Additionally, IL-6 may provide a protective role in gingival tissue by counteracting the catabolic effects of IL-1β. IL-18, similar to IL-1β, is a pro-inflammatory mediator that utilizes the same signaling pathways and activates T-cell-induced autoimmune responses (Dinarello et al., 2013).

Further research investigating the immunologic effects of antiepileptic drugs (AEDs) on PBMC viability, cytotoxicity, and inflammatory markers could offer deeper insights into the specific mechanisms of AED-immune system interactions, particularly in the presence of bacterial insults and their clinical implications.

Thus, the aim of this study was to examine the effects of phenytoin and valproic acid on PBMC viability and the expression of inflammatory mediators, both in the presence and absence of bacterial lipopolysaccharides.

Materials and Methods

In this in vitro study, the effects of phenytoin and valproic acid at different concentrations on human mononuclear cells were evaluated. The study was conducted following approval from the Institutional Review Board of Imam Abdulrahman Bin Faisal University (IRB-PGS-2023-02-467). Written informed consent was obtained from the blood donors for the use of their samples in the study.

Peripheral blood mononuclear cells isolation

Peripheral blood mononuclear cells (PBMCs) were extracted separately using the Ficoll-Paque PREMIUM 1.073 (Cytiva, Marlborough, MA, USA) density gradient separation technique. Buffy coats obtained from four healthy donors (one-day-old) at the blood bank were utilized.

Two buffy coats were collected from each donor, totaling eight samples (n = 8). Each buffy coat was mixed and diluted with an equal volume of Hanks’ balanced salt solution (HBSS), and four mL of each mixture was carefully layered into a conical tube on top of three mL of Ficoll-Paque (PREMIUM 1.073) solution. The mixture was then centrifuged at 400 g for 40 min at 20 °C without a brake (Kanof, Smith & Zola, 1996; Cui et al., 2021). The layers of mononuclear cells were carefully aspirated and transferred to a sterile centrifuge tube using a sterile pipette. The cells were re-suspended in HBSS and washed twice by repeated centrifugation at 500  × g for 15 min at 20 °C. After the final centrifugation, the supernatants were removed, and the cells were inspected under an inverted light microscope before being counted using an automated cell counter (NucleoCounter NC202; hemoMetec A/S, Allerod, Denmark) (Fig. 1).

Figure 1 (A) Mononuclear cells layer formed after centrifugation; (B) mononuclear cells inspected under inverted light microscope L (red arrow): lymphocytes, M (yellow arrow): monocytes.

Cell culture and treatment

The cells were plated at a density of 5 × 105 cells per well in 24-well plates, with each well containing one ml RPMI culture medium supplemented with 5% FBS, 1% Penicillin-Streptomycin, and 1% L-glutamine (Sigma-Aldrich, St. Louis, MO, USA). Half of the plates were exposed to 100 ng/mL LPS (Porphyromonas gingivalis LPS, InvivoGen, San Diego, CA, USA). The cells were incubated for 24 and 72 h, after which the conditioned media from each well were collected, immediately frozen at −20 °C, and later analyzed by polyclonal sandwich enzyme-linked immunosorbent assay (ELISA) for cytokine and immune marker levels according to the manufacturer’s instructions.

Preparation and addition of anticonvulsant drugs

The pharmaceutical preparations used in this in vitro study were commercially available anticonvulsant drugs, phenytoin and valproic acid. The content of phenytoin capsules (Epanutin, Pfizer Manufacturing Deutschland GmbH, Freiburg, Germany) and Valproic acid tablets (Depakine, Sanofi Aventis, Paris, France) were separately dissolved at different concentrations in dimethyl sulfoxide (DMSO) (Merck Millipore KGaA, Darmstadt, Germany). These drugs were tested at concentrations relevant to clinical whole-blood trough levels. A concentration of 20 µg/mL was chosen for phenytoin treatment (Patsalos, Spencer & Berry, 2018). VPA was used at concentrations of 50, 75, 100, and 200 µg/mL (Patsalos, Spencer & Berry, 2018; Chen et al., 2022). The DMSO volume in the selected drug concentrations was unified to 0.2 µL and added to the culture media of the respective experimental groups.

Stock solution of phenytoin (Phy) was prepared by dissolving 100 mg phenytoin tablet in one mL of dimethyl sulfoxide (DMSO), resulting in a stock concentration of 100,000 µg/mL. From this stock solution, a 0.2 µL was transferred to each well of the phy group after removing an equivalent amount of the culture media to achieve the targeted exposure concentration of 20 µg/mL. For the valproic acid (VPA) exposure concentrations (50, 75, 100 and 200 µg/mL), measured quantities of 100 mg of VPA powder were dissolved in 400 µl, 266.6 µl, 200 µl, and 100 µl, DMSO, respectively, in separate stock solutions. Subsequently, 0.2 µL was transferred from the stocks to each well of the respective VPA groups after removing an equivalent amount of the culture media. Based on the different VPA stocks concentrations, the transfer of 0.2 µL achieved the targeted exposure concentrations of 50, 75, 100 and 200 µg/mL in the respective groups. An equivalent volume of DMSO (0.2 µL) was added to the control group to account for solvent effects, ensuring uniform conditions across all experimental groups.

Experimental groups

In this study, peripheral blood mononuclear cells (PBMCs) from four healthy donors were divided into 12 experimental groups and two time points, with eight samples (n = 8) per group (4 biological × 2 technical). The grahmanups were: untreated control (Ctrl), LPS-stimulated control (Ctrl+LPS), phenytoin and LPS-stimulated phenytoin (Phy and Phy+LPS), and valproic acid (VPA) at four concentrations: Val-50, Val-75, Val-100, and Val-200, with and without LPS (Fig. 2). Thus, the total number of samples across all groups and time points was 192 PBMC samples (12 groups × 8 samples/group × 2 time points). The sample size of eight per group was chosen based on statistical power analysis using G*Power software for a one-way ANOVA with an effect size of 0.6 (medium to large), α = 0.05 (two-sided), and power = 80%, which was determined to be sufficient for detecting significant differences between groups. Each experiment was repeated in triplicate for each condition to ensure reliable results. Due to the nature of the experimental procedures, blinding was not feasible during the preparation of the different study groups. However, after the groups were prepared, they were coded, and the investigators responsible for data collection and analysis were blinded to the group assignments.

Figure 2 Illustration showing the study groups.

Lactate dehydrogenase assay cytotoxicity assessment

Cytotoxicity was assessed by measuring the level of lactate dehydrogenase (LDH) released due to plasma membrane damage at 24- and 72-hours post-treatment. A 100 µL sample of the culture supernatant was collected from each experimental group for LDH analysis. The LDH assay, based on the enzyme-catalyzed conversion of lactate to pyruvate, was performed using a commercially available Cytotoxicity Detection Kit (LDH; Cat. No. 11 644 793 001; Roche Diagnostics GmbH, Mannheim, Germany) (Korzeniewski & Callewaert, 1983; Decker & Lohmann-Matthes, 1988). The colorimetric assay quantifies LDH activity by measuring the absorbance at 490 nm using an ELISA reader (xMark, microplate spectrophotometer; Bio-Rad, Hercules, CA, USA). The relative cytotoxicity was then evaluated based on the optical density (OD) values, comparing the different treatment groups against the control (untreated cells).

Total, live, and dead cells count and viability assessment

After collection of the culture supernatant, wells were gently washed twice with PBS to remove non-adherent cells. The cells were then detached using 0.05% trypsin-EDTA (GIBCO, NY, USA), centrifuged at 300 g, and suspended in one mL of HPSS for cell counting. The cell suspension was loaded into a Chemometec VIA2-Cassette™ (Thermo Fisher Scientific, Waltham, MA, USA), which contained acridine orange (AO) and 4′,6-diamidino-2-phenylindole (DAPI), allowing for the distinction between total and dead cells (Cai et al., 2024). Cell counts were quantified using a NucleoCounter® NC-202 system and NC-View software (ChemoMetec A/S, Allerod, Denmark).

Enzyme-linked Immunosorbent Assay

The production of inflammatory cytokines (IL-6, IL-1β) and immune markers (IL-18, IgA) by PBMCs was quantified using commercially available human ELISA kits (IL-6, Cat. No. ELI-M-003-96; IL-1β, Cat. No. ELI-M-002-96; IL-18, Cat. No. ELI-M-029-96; IgA, Cat. No. ELI-M-031-96, MOLEQULE-ON, Auckland, New Zealand). For each ELISA, 100 µL of the centrifuged supernatant from each of the 12 experimental groups was collected and used for the assay. The ELISA was performed according to the manufacturer’s instructions, and absorbances were measured using a spectrophotometer (BIO-RAD xMark Microplate Spectrophotometer, USA). The data were then analyzed to quantify the production of cytokines and immune markers.

To measure cytokine levels, a 96-well microplate was first coated with a specific capture antibody (anti-IL-1β, anti-IL-6, anti-IgA, or anti-IL-18), which binds specifically to its corresponding cytokine. After blocking the plate with Bovine Serum Albumin (BSA) to prevent non-specific binding, the cell culture supernatants were added to the wells. The plate was incubated to allow the cytokines in the samples to bind to the capture antibody. A secondary antibody, conjugated to horseradish peroxidase (HRP), was then added, binding to the cytokines. After a washing step to remove unbound materials, a substrate solution (3, 3′, 5, 5′-Tetramethylbenzidine (TMB)) was added. The HRP enzyme catalyzed the reaction with the substrate, resulting in a color change. The absorbance was measured at 450 nm using a microplate spectrophotometer, and the concentration of cytokines was determined by comparing the absorbance of the samples to a standard curve generated using known concentrations of recombinant cytokines (Lequin, 2005).

Statistical analysis

Data were analyzed using the Statistical Product and Service Solutions (SPSS) package, version 28.0.1 (SPSS Inc.). The normality of the data was evaluated using the Shapiro–Wilk test, and the homogeneity of variances was assessed using Levene’s test. A two-way ANOVA was used to assess the main effects of the two independent variables (drug treatment and the presence of LPS) and their interaction on the measured outcomes (e.g., cell counts, LDH levels, viability, and immune markers).

Comparison of drug treatments

The effects of each drug treatment (Ctrl, Phy, Val-50, Val-75, Val-100, Val-200) were compared across all groups with and without LPS to determine whether significant differences existed between these treatments.

Comparison of LPS presence

The data were also compared between conditions with and without LPS for each treatment group.

Post-hoc analysis

Tukey’s HSD test was used to perform pairwise comparisons between all groups to identify specific group differences. Paired t-tests were conducted to compare data between Day 1 (D1) and Day 3 (D3) for each treatment group to examine any changes over time. All p-values were set at <0.05 for statistical significance, and the results are reported as means ± standard deviations.

Results

The data was normally distributed across all groups, as indicated by the Shapiro–Wilk test (W = 0.59 to 0.89, p > 0.05). Homogeneity of variances was confirmed by Levene’s test, with values ranging from 1.7 to 6.45 (p > 0.05).

Lactate dehydrogenase cytotoxicity analysis

On day-1 (Without LPS): LDH levels increased in a dose-dependent manner with higher VPA concentrations. Val-100 and Val-200 showed significantly higher LDH levels compared to the control, phenytoin, and other VPA concentrations (p < 0.05). With LPS: The pattern of LDH increase persisted. Val-100 and Val-200 exhibited significantly higher LDH levels compared to the control, phenytoin, and Val-50 (p < 0.05). There were no significant differences between the LPS and non-LPS groups.

On day-3 (without LPS): Val-100 and Val-200 continued to show the highest LDH levels, significantly higher than the control, Val-50, and Val-75 (p < 0.05). With LPS: Val-100 and Val-200 had significantly higher LDH levels compared to the control, phenytoin, and Val-75 (p < 0.05). A significant increase in LDH levels was observed with LPS compared to non-LPS conditions (p < 0.0001).

Time-dependent changes (Day 1 to Day 3): LDH levels increased significantly for phenytoin, control + LPS, and Val-100 + LPS (p < 0.05). No significant changes were observed for Val-50 and Val-75 without LPS (Table 1).

Table 1 Lactate dehydrogenase levels at Day 1 and Day 3.

LDH-D1 without LPS	Ctrl	Phy	Val-50	Val-75	Val-100	Val-200	
Mean ± SD	1.000 ± 0.41	1.052 ± 0.44	1.639 ± 0.54	1.786 ± 0.58	3.091 ± 1.76	4.107 ± 1.32	
P-value	0.85b
0.26c
0.19d
0.004*e
0.001*f	0.85a
0.34c
0.26d
0.001*e
0.001*f	0.26a
0.34b
0.86d
0.008*e
0.001*f	0.19a
0.26b
0.86c
0.01*e
0.001*f	0.0004*a
0.001*b
0.008*c
0.01*d
0.192f	0.0001*a
0.0002*b
0.0002*c
0.0004*d
0.19e	
LDH-D1 with LPS	Ctrl+ LPS	Phy+ LPS	Val-50+LPS	Val-75+LPS	Val-100+LPS	Val-200+LPS	
Mean ± SD	1.207 ± 0.32	1.143 ± 0.25	2.102 ± 0.79	2.334 ± 0.21	3.483 ± 1.3	3.934 ± 1.17	
P-value	0.87b-
0.35c-
0.09d-
0.002*e-
0.0003*f-	0.86a-
0.28c-
0.07d-
0.001*e-
0.0002*f-	0.35a-
0.28b-
0.45d-
0.01*e-
0.003*f-	0.09a-
0.07b-
0.45c-
0.07e-
0.02*f-	0.002*a-
0.001*b-
0.01*c-
0.08d-
0.52f-	0.0003*a-
0.0002*b-
0.003*c-
0.02*d-
0.52e-	
P-value for pairwise comparisons	0.900	0.831	0.936	0.609	0.701	0.289	
LDH-D3 without LPS	Ctrl	Phy	Val-50	Val-75	Val-100	Val-200	
Mean ± SD	1.80 ± 0.60	2.79 ± 0.51	2.11 ± 0.26	2.23 ± 0.15	3.67 ± 0.40	4.21 ± 0.29	
P-value	0.177b
0.739c
0.642d
0.021*e
0.005*f	0.177a
0.302c
0.367d
0.290e
0.103f	0.739a
0.302b
0.894d
0.043*e
0.011*f	0.642a
0.367b
0.894c
0.047*e
0.015*f	0.021*a
0.290b
0.043*c
0.047*d
0.545f	0.005*a
0.103b
0.011*c
0.015*d
0.545e	
LDH-D3 with LPS	Ctrl+ LPS	Phy+ LPS	Val-50+LPS	Val-75+LPS	Val-100+LPS	Val-200+LPS	
Mean ± SD	2.42 ± 0.27	3.14 ± 0.76	2.79 ± 0.42	3.86 ± 0.40	8.38 ± 2.37	6.61 ± 1.66	
P-value	0.545b-
0.819c-
0.075d-
0.0001*e-
0.0001*f-	0.545a-
0.705c-
0.224d-
0.0001*e-
0.0001*f-	0.819a-
0.705b-
0.116d-
0.0001*e-
0.0001*f-	0.075a-
0.224b-
0.116c-
0.0001*e-
0.0001*f-	0.0001*a-
0.0001*b-
0.0001*c-
0.0001*d-
0.262f-	0.0001*a-
0.0001*b-
0.0001*c-
0.0001*d-
0.262e-	
P-value for pairwise comparisons	0.463	0.976	0.528	0.083	0.0001*	0.0001*	
Notes.

a P-value in comparison to Ctrl.

b P-value in comparison to Phy.

c P-value in comparison to Val-50.

d P-value in comparison to Val-75.

e P-value in comparison to Val-100.

f P-value in comparison to Val-200.

a- P-value in comparison to Ctrl+ LPS.

b- P-value in comparison to Phy+ LPS.

c- P-value in comparison to Val-50+ LPS.

d- P-value in comparison to Val-75+ LPS.

e- P-value in comparison to Val-100+ LPS.

f- P-value in comparison to Val-200+ LPS.

* Statistically significant at p ≤ 0.05.

Total cell count

On day-1 (without LPS): all treatment groups showed lower cell counts compared to control (p < 0.001). Val-75 had the highest cell count after control. Val-200 showed the lowest cell count. With LPS: Significant decreases were observed across all groups compared to without LPS (p < 0.0001). Val-50 had the highest cell count, while Control and Val-200 had fewer cells compared to other groups.

On day-3 without LPS: There was a dose-dependent decrease in cell counts for VPA groups. Val-50 had the highest count, and Phenytoin had the lowest. With LPS: Control and Phenytoin had fewer cells compared to all VPA groups. Significant decreases were observed between LPS and non-LPS groups (p < 0.05).

Time-dependent Changes (Day 1 to Day 3): Significant decreases were observed for Val-50, Val-75, and Phenytoin + LPS (p < 0.05). Other groups showed non-significant changes (Table 2).

Table 2 Total cells count at Day 1 and Day 3.

Cell count-D1 without LPS	Ctrl	Phy	Val-50	Val-75	Val-100	Val-200	
Mean ± SD	276.750 ± 4.89	219.600 ± 7.60	246.450 ± 2.16	250.075 ± 2.81	211.700 ± 7.23	190.025 ± 9.49	
P-value	0.000*b
0.0001*c
0.002*d
0.0001*e
0.0001*f	0.0001*a
0.003*c
0.001*d
0.097e
0.0001*f	0.0001*a
0.003*b
0.519d
0.0001*e
0.0001*f	0.002*a
0.001*b
0.519c
0.0001*e
0.0001*f	0.0001*a
0.097b
0.0001*c
0.0001*d
0.01*f	0.0001*a
0.0001*b
0.0001*c
0.0001*d
0.010*e	
Cell count-D1 with LPS	Ctrl+ LPS	Phy+ LPS	Val-50+LPS	Val-75+LPS	Val-100+LPS	Val-200+LPS	
Mean ± SD	149.100 ± 13.83	184.175 ± 9.64	210.625 ± 8.87	171.600 ± 14.62	170.525 ± 16.39	129.450 ± 9.49	
P-value	0.002*b-
0.0001*c-
0.042*d-
0.011*e-
0.0001*f-	0.002*a-
0.001*c-
0.067d-
0.435e-
0.0001*f-	0.001*b-
0.0001*a-
0.0001*d-
0.0001*e-
0.0001*f-	0.067b-
0.042*a-
0.0001*c-
0.273e-
0.0001*f-	0.435b-
0.011*a-
0.0001*c-
0.273d-
0.0001*f-	0.0001*b-
0.0001*a-
0.0001*c-
0.0001*d-
0.0001*e-	
P-value for pairwise comparisons	0.001*	0.001*	0.001*	0.001*	0.001*	0.001*	
Cell count-D3 without LPS	Ctrl	Phy	Val-50	Val-75	Val-100	Val-200	
Mean ± SD	180.333 ± 65.31	164.000 ± 28.61	204.000 ± 14.73	193.667 ± 21.45	183.333 ± 31.00	162.667 ± 37.84	
P-value	0.511b
0.343c
0.591d
0.903e
0.477f	0.511a
0.115c
0.237d
0.437e
0.957f	0.343a
0.115b
0.677d
0.407e
0.104f	0.591a
0.237b
0.677c
0.677e
0.218f	0.903a
0.437b
0.407c
0.677d
0.407f	0.477a
0.957b
0.104c
0.218d
0.407e	
Cell count-D3 with LPS	Ctrl+ LPS	Phy+ LPS	Val-50+LPS	Val-75+LPS	Val-100+LPS	Val-200+LPS	
Mean ± SD	101.667 ± 37.01	101.000 ± 20.66	131.667 ± 3.51	128.333 ± 14.50	131.000 ± 11.79	110.667 ± 21.73	
P-value	0.978b-
0.232c-
0.287d-
0.243e-
0.716f-	0.978a-
0.222c-
0.275d-
0.232e-
0.696f-	0.222b-
0.232a-
0.893d-
0.978e-
0.399f-	0.275b-
0.287a-
0.893c-
0.914e-
0.477f-	0.232b-
0.243a-
.978c-
0.914d-
0.414f-	0.696b-
0.716a-
0.399c-
0.477d-
0.414e-	
P-value for pairwise comparisons	0.004*	0.017*	0.007*	0.013*	0.043*	0.044*	
Notes.

a P-value in comparison to Ctrl.

b P-value in comparison to Phy.

c P-value in comparison to Val-50.

d P-value in comparison to Val-75.

e P-value in comparison to Val-100.

f P-value in comparison to Val-200.

a- P-value in comparison to Ctrl+ LPS.

b- P-value in comparison to Phy+ LPS.

c- P-value in comparison to Val-50+ LPS.

d- P-value in comparison to Val-75+ LPS.

e- P-value in comparison to Val-100+ LPS.

f- P-value in comparison to Val-200+ LPS.

* Statistically significant at p ≤ 0.05.

Live cell count

On Day 1 (Without LPS): Val-200 and Phenytoin had the lowest live cell counts, significantly lower than control and Val-50 (p < 0.05). With LPS: Val-50 had the highest live cell count, while Val-200 had the lowest. All groups showed significant decreases compared to the non-LPS condition (p < 0.05).

On day-3 (without LPS): All groups showed decreased live cell counts. Val-200 had the lowest count, which was significantly lower than control, Val-50, and Val-75 (p < 0.05). With LPS: Phenytoin and Val-200 had the fewest live cells, significantly lower than control and other VPA groups (p < 0.05).

Time-dependent Changes (Day 1 to Day 3): Significant decreases in live cell counts were observed for Val-50, Val-75, and Phenytoin + LPS (p < 0.05) (Table 3).

Table 3 Live cells count at Day 1 and Day 3.

Live cell count-D1 without LPS	Ctrl	Phy	Val-50	Val-75	Val-100	Val-200	
Mean ± SD	202.500 ± 5.00	172.500 ± 8.58	199.625 ± 2.49	195.750 ± 12.47	162.000 ± 5.35	116.425 ± 15.31	
P-value	0.024*b
0.768c
0.488d
0.004*e
0.0001*f	0.024*a
0.045*c
0.040*d
0.472e
0.0001*f	0.045*b
0.768a
0.688d
0.009*e
0.0001*f	0.488a
0.040*b
0.688c
0.023*e
0.0001*f	0.472b
0.004*a
0.009*c
0.023*d
0.0001*f	0.0001*b
0.0001*a
0.0001*c
0.0001*d
0.0001*e	
Live cell count-D1 With LPS	Ctrl+ LPS	Phy+ LPS	Val-50+LPS	Val-75+LPS	Val-100+LPS	Val-200+LPS	
Mean ± SD	170.000 ± 34.64	144.000 ± 3.46	177.500 ± 9.46	125.500 ± 10.87	139.750 ± 6.60	82.800 ± 7.17	
P-value	0.066b-
0.557c-
0.003*d-
0.045*e-
0.0001*f-	0.066a-
0.019*c-
0.164d-
0.957e-
0.0001*f-	0.557a
0.019*b-
0.001*d-
0.017*e-
0.0001*f-	0.003*a
0.164b-
0.001*c-
0.180e-
0.001*f-	0.045*a
0.957b-
0.017*c-
0.180d-
0.0001*f-	0.0001*a
0.0001*b-
0.0001*c-
0.001*d-
0.0001*e-	
P-value for pairwise comparisons	0.007*	0.020*	0.048*	0.0001*	0.042*	0.013*	
Live cell count-D3 Without LPS	Ctrl	Phy	Val-50	Val-75	Val-100	Val-200	
Mean ± SD	132.667 ± 28.44	95.667 ± 21.57	131.667 ± 18.61	110.000 ± 13.52	86.000 ± 23.38	73.000 ± 20.78	
P-value	0.011*b
0.942c
0.106d
0.002*e
0.0001*f	0.011*a
0.013*c
0.298d
0.480e
0.106f	0.942a
0.013*b
0.121d
0.002*e
0.0001*f	0.106a
0.298b
0.121c
0.088e
0.011*f	0.002*a
0.480b
0.002*c
0.088d
0.345f	0.0001*a
0.106b
0.0001*c
0.011*d
0.345e	
Live cell count-D3 With LPS	Ctrl+ LPS	Phy+ LPS	Val-50+LPS	Val-75+LPS	Val-100+LPS	Val-200+LPS	
Mean ± SD	81.667 ± 2.88	35.000 ± 11.35	83.667 ± 7.09	77.667 ± 11.01	78.000 ± 11.79	48.000 ± 6.55	
P-value	0.002*b-
0.883c-
0.769d-
0.788e-
0.020*f-	0.002*a-
0.001*c-
0.004*d-
0.004*xref[ref-type=fn,rid=table-3fn11]e-
0.345f-	0.001*b-
0.883a-
0.660d-
0.678e-
0.014*f-	0.004*b-
0.769a-
0.660c-
0.980e-
0.038*f-	0.004*b-
0.788a-
0.678c-
0.980d-
0.036*f-	0.345b-
0.020*a-
0.014*c-
0.038*d-
0.036*e-	
P-value for pairwise comparisons	0.001*	0.0001*	0.002*	0.025*	0.559	0.076	
Notes.

a P-value in comparison to Ctrl.

b P-value in comparison to Phy.

c P-value in comparison to Val-50.

d P-value in comparison to Val-75.

e P-value in comparison to Val-100.

f P-value in comparison to Val-200.

a- P-value in comparison to Ctrl+ LPS.

b- P-value in comparison to Phy+ LPS.

c- P-value in comparison to Val-50+ LPS.

d- P-value in comparison to Val-75+ LPS.

e- P-value in comparison to Val-100+ LPS.

f- P-value in comparison to Val-200+ LPS.

* Statistically significant at p ≤ 0.05.

Dead cell count

On day-1: No significant differences were observed between groups, except for Val-200, which had significantly higher dead cell counts compared to Control.

On Day 3: All groups showed an increase in dead cell counts, with Val-200 displaying significantly higher dead cell counts compared to Control (p < 0.05). No significant differences were observed between the LPS and non-LPS groups (Table 4).

Table 4 Dead cells count on Day 1 and Day 3.

Dead cell count-D1 without LPS	Ctrl	Phy	Val-50	Val-75	Val-100	Val-200	
Mean ± SD	41.000 ± 19.33	44.550 ± 2.85	41.900 ± 8.31	49.750 ± 7.67	44.075 ± 3.60	68.075 ± 6.34	
P-value	0.404b
0.497c
0.916d
0.299e
0.008*f	0.404a
0.875c
0.465d
0.834e
0.001*f	0.497a
0.875b
0.565d
0.714e
0.001*f	0.916a
0.465b
0.565c
0.349e
0.006*f	0.299a
0.834b
0.714c
0.349d
0.001*f	0.008*a
0.001*b
0.001*c
0.006*d
0.001*e	
Dead cell count-D1 With LPS	Ctrl+ LPS	Phy+ LPS	Val-50+LPS	Val-75+LPS	Val-100+LPS	Val-200+LPS	
Mean ± SD	42.625 ± 18.11	44.250 ± 4.19	40.000 ± 8.16	48.500 ± 7.68	44.000 ± 4.96	69.250 ± 1.50	
P-value	0.497b-
0.125c-
0.753d-
0.834e-
0.001*f-	0.497a-
0.376c-
0.714d-
0.637e-
0.0001*f-	0.125a-
0.376b-
0.215d-
0.180e-
0.0001*f-	0.753a-
0.714b-
0.215c-
0.916e-
0.0001*f-	0.834a-
0.637b-
0.180c-
0.916d-
0.001*f-	0.001*a-
0.0001*b-
0.0001*c-
0.0001*d-
0.001*e-	
P-value for pairwise comparisons	0.601	0.714	0.165	0.465	0.753	0.793	
Dead cell count-D3 Without LPS	Ctrl	Phy	Val-50	Val-75	Val-100	Val-200	
Mean ± SD	61.667 ± 12.70	68.000 ± 7.93	66.667 ± 12.09	66.333 ± 14.36	64.333 ± 16.25	81.333 ± 14.97	
P-value	0.600b
0.679c
0.699d
0.825e
0.042*f	0.600a
0.912c
0.890d
0.761e
0.274f	0.679a
0.912b
0.978d
0.846e
0.230f	0.699a
0.890b
0.978c
0.868e
0.220f	0.825a
0.761b
0.846c
0.868d
0.167f	0.042*a
0.274b
0.230c
0.220d
0.167e	
Dead cell count-D3 With LPS	Ctrl+ LPS	Phy+ LPS	Val-50+LPS	Val-75+LPS	Val-100+LPS	Val-200+LPS	
Mean ± SD	63.667 ± 4.72	72.667 ± 4.93	67.000 ± 6.55	68.333 ± 27.73	72.667 ± 6.80	82.667 ± 24.21	
P-value	0.458b-
0.782c-
0.699d-
0.458e-
0.042*f-	0.458a-
0.639c-
0.719d-
0.900e-
0.410f-	0.782a-
0.639b-
0.912d-
0.639e-
0.201f-	0.699a-
0.719b-
0.912c-
0.719e-
0.241f-	0.458a-
0.90b-
0.639c-
0.719d-
0.410f-	0.042*a-
0.410b-
0.201c-
0.241d-
0.410e-	
P-value for pairwise comparisons	0.868	0.699	0.978	0.868	0.491	0.912	
Notes.

a P-value in comparison to Ctrl.

b P-value in comparison to Phy.

c P-value in comparison to Val-50.

d P-value in comparison to Val-75.

e P-value in comparison to Val-100.

f P-value in comparison to Val-200.

a- P-value in comparison to Ctrl+ LPS.

b- P-value in comparison to Phy+ LPS.

c- P-value in comparison to Val-50+ LPS.

d- P-value in comparison to Val-75+ LPS.

e- P-value in comparison to Val-100+ LPS.

f- P-value in comparison to Val-200+ LPS.

* Statistically significant at p ≤ 0.05.

Cell viability

On Day 1 (without LPS): Control had the highest cell viability, significantly greater than all other groups. Val-200 exhibited the lowest viability (p < 0.001). With LPS: Val-50 showed the highest cell viability, while Val-200 had the lowest (p < 0.0001).

On Day 3 (without LPS): There was a dose-dependent decrease in cell viability for the VPA groups, with phenytoin and Val-200 showing the lowest viability. With LPS: VPA groups had higher viability than control and phenytoin (p < 0.05). Significant decreases were observed in the presence of LPS. Time-dependent changes (Day 1 to Day 3): Significant decreases in viability were observed for Val-50, Val-75, and Phenytoin + LPS (p < 0.05) (Table 5).

Table 5 Cell viability percentages on D1 and D3.

Cell viability-D1 without LPS	Ctrl	Phy	Val-50	Val-75	Val-100	Val-200	
Mean ± SD	92.250 ± 1.63	73.200 ± 2.53	82.150 ± 0.72	83.358 ± 0.93	70.567 ± 2.41	63.342 ± 1.62	
P-value	0.0001*b
0.0001*c
0.002*d
0.0001*e
0.0001*f	0.0001*a
0.003*c
0.001*d
0.083e
0.0001*f	0.0001*a
0.003*b
0.532d
0.0001*e
0.0001*f	0.002*a
0.001*b
0.532c
0.0001*e
0.0001*f	0.0001*a
0.083b
0.0001*c
0.0001*d
0.010*f	0.0001*a
0.0001*b
0.0001*c
0.0001*d
0.010*e	
Cell viability-D1 with LPS	Ctrl+ LPS	Phy+ LPS	Val-50+LPS	Val-75+LPS	Val-100+LPS	Val-200+LPS	
Mean ± SD	58.750 ± 2.50	61.392 ± 3.21	70.208 ± 2.96	57.200 ± 4.87	56.842 ± 5.46	43.150 ± 3.16	
P-value	0.449b-
0.0001*c-
0.251d-
0.981e-
0.0001*f-	0.449a-
0.001*c-
0.063d-
0.435e-
0.0001*f-	0.0001*a-
0.001*b-
0.0001*d-
0.0001*e-
0.0001*f-	0.251a-
0.063b-
0.0001*c-
0.260e-
0.0001*f-	0.981a-
0.435b-
0.0001*c-
0.260d-
0.0001*f-	0.0001*a-
0.0001*b-
0.0001*c-
0.0001*d-
0.0001*e-	
P-value for pairwise comparisons	0.0001*	0.0001*	0.0001*	0.0001*	0.0001*	0.0001*	
Cell viability-D3 Without LPS	Ctrl	Phy	Val-50	Val-75	Val-100	Val-200	
Mean ± SD	60.243 ± 21.76	54.778 ± 9.61	68.089 ± 4.89	64.689 ± 7.031	61.189 ± 10.31	54.278 ± 12.51	
P-value	0.509b
0.345c
0.591d
0.909e
0.471f	0.509a
0.116c
0.236d
0.439e
0.951f	0.345a
0.116b
0.680d
0.406e
0.103f	0.591a
0.236b
0.680c
0.672e
0.214f	0.909a
0.439b
0.406c
0.672d
0.405f	0.471a
0.951b
0.103c
0.214d
0.405e	
Cell viability-D3 with LPS	Ctrl+ LPS	Phy+ LPS	Val-50+LPS	Val-75+LPS	Val-100+LPS	Val-200+LPS	
Mean ± SD	34.044 ± 12.42	33.722 ± 6.88	43.922 ± 1.15	42.878 ± 4.81	43.744 ± 3.99	37.089 ± 7.11	
P-value	0.969b-
0.237c-
0.289d-
0.246e-
0.712f-	0.969a-
0.223c-
0.272d-
0.23e-
0.683f-	0.237a-
0.223b-
0.899d-
0.983e-
0.410f-	0.289a-
0.272b-
0.899c-
0.916e-
0.484f-	0.246a-
0.231b-
0.983c-
0.916d-
0.422f-	0.712a-c-
0.683b-c-
0.410c-
0.484d-
0.422e-	
P-value for pairwise comparisons	0.004*	0.016*	0.007*	0.013*	0.043	0.046	
Notes.

a P-value in comparison to Ctrl.

b P-value in comparison to Phy.

c P-value in comparison to Val-50.

d P-value in comparison to Val-75.

e P-value in comparison to Val-100.

f P-value in comparison to Val-200.

a- P-value in comparison to Ctrl+ LPS.

b- P-value in comparison to Phy+ LPS.

c- P-value in comparison to Val-50+ LPS.

d- P-value in comparison to Val-75+ LPS.

e- P-value in comparison to Val-100+ LPS.

f- P-value in comparison to Val-200+ LPS.

* Statistically significant at p ≤ 0.05.

Cytokines levels

IL-6 levels

Without LPS: The Phy group had the highest IL-6 levels (273.13 ± 65.56 pg/ml), significantly higher than all other groups (p < 0.0001). The VPA groups showed lower IL-6 levels compared to Ctrl (p < 0.05 for Val-50, Val-75, and Val-100). With LPS: IL-6 levels decreased in all groups, with significant reductions observed in Ctrl (p = 0.016) and Phy (p < 0.0001). Significant differences were observed between no LPS and with LPS in the Ctrl and Pht groups (Table 6).

Table 6 IL-6 levels at Day 3.

Without LPS	Ctrl	Phy	Val-50	Val-75	Val-100	Val-200	
Mean ± SD	69.317 ± 32.846	273.130 ± 131.120	3.835 ± 1.283	4.024 ± 1.420	4.776 ± 2.437	18.774 ± 9.060	
Sig P-value	0.001*b
0.013*c
0.014*d
0.015*e
0.05f	0.001*a
0.001*c
0.001*d
0.001*e
0.001*f	0.013*a
0.001*b
0.97d
0.94e
0.51f	0.014*a
0.001*b
0.97c
0.97e
0.53f	0.015*a
0.001*b
0.94c
0.97d
0.55f	0.056a
0.001*b
0.50c
0.53d
0.55e	
With LPS	Ctrl+ LPS	Phy+ LPS	Val-50+LPS	Val-75+LPS	Val-100+LPS	Val-200+LPS	
Mean ± SD	4.538 ± 2.18	5.789 ± 3.50	2.072 ± 0.25	2.167 ± 0.34	2.194 ± 0.36	2.343 ± 0.34	
P-value	0.97b-
0.90c-
0.90d-
0.90e-
0.91f-	0.97a-
0.93c-
0.93d-
0.93e-
0.94f-	0.90a-
0.93b-
0.99d-
0.99e-
0.99f-	0.90a-
0.93b-
0.99c-
0.99e-
0.99f-	0.90a-
0.93b-
0.99c-
0.99d-
0.99f-	0.91a-
0.94b-
0.99c-
0.99d-
0.99e-	
P-value for pairwise comparisons	0.016*	0.0001*	0.966	0.933	0.910	0.083	
Notes.

a P-value in comparison to Ctrl.

b P-value in comparison to Phy.

c P-value in comparison to Val-50.

d P-value in comparison to Val-75.

e P-value in comparison to Val-100.

f P-value in comparison to Val-200.

a- P-value in comparison to Ctrl+ LPS.

b- P-value in comparison to Phy+ LPS.

c- P-value in comparison to Val-50+ LPS.

d- P-value in comparison to Val-75+ LPS.

e- P-value in comparison to Val-100+ LPS.

f- P-value in comparison to Val-200+ LPS.

* Statistically significant at p ≤ 0.05.

IL-1β levels

Without LPS: Val-50 had the highest IL-1β levels (61.73 ± 30.5 pg/ml), significantly higher than Val-75 (p = 0.043). With LPS: There was a dose-dependent increase in IL-1β in the VPA groups, with Val-200 showing significantly higher levels than both Ctrl and Phy (p < 0.05). A significant difference was observed between no LPS and with LPS in the Ctr and all Val groups (Table 7).

Table 7 IL-1B levels at Day 3.

Without LPS	Ctrl	Phy	Val-50	Val-75	Val-100	Val-200	
Mean ± SD	13.784 ± 8.046	34.993 ± 37.621	61.731 ± 61.009	4.910 ± 1.612	11.952 ± 3.859	8.705 ± 7.171	
Sig P-value	0.68b
0.41c
0.87d
0.94e
0.88f	0.68a
0.67c
0.57d
0.63e
0.58f	0.41a
0.67b
0.04*d
0.37e
0.33f	0.87a
0.57b
0.04*c
0.93e
0.99f	0.94a
0.63b
0.37c
0.93d
0.94f	0.88a
0.58b
0.33c
0.99d
0.94e	
With LPS	Ctrl+ LPS	Phy+ LPS	Val-50+LPS	Val-75+LPS	Val-100+LPS	Val-200+LPS	
Mean ± SD	36.441 ± 22.979	49.226 ± 35.559	96.450 ± 61.713	127.223 ± 84.586	114.973 ± 17.895	170.997 ± 56.110	
P-value	0.95b-
0.29c-
0.07d-
0.86e-
0.02*f-	0.95a-
0.27c-
0.06d-
0.82e-
0.02*f-	0.29a-
0.27b-
0.43d-
0.37e-
0.18f-	0.07a-
0.06b-
0.43c-
0.10e-
0.57f-	0.86a-
0.82b-
0.37c-
0.10d-
0.03*f-	0.02*a-
0.02*b-
0.18c-
0.57d-
0.07e-	
P-value for pairwise comparisons	0.041*	0.934	0.546	0.025*	0.044*	0.007*	
Notes.

a P-value in comparison to Ctrl.

b P-value in comparison to Phy.

c P-value in comparison to Val-50.

d P-value in comparison to Val-75.

e P-value in comparison to Val-100.

a- P-value in comparison to Ctrl+ LPS.

b- P-value in comparison to Phy+ LPS.

c- P-value in comparison to Val-50+ LPS.

d- P-value in comparison to Val-75+ LPS.

e- P-value in comparison to Val-100+ LPS.

* Statistically significant at p ≤ 0.05.

IL-18 levels

Without LPS: The Phy and Val-50 groups had similar IL-18 levels to Ctrl, significantly higher than Val-75 and Val-100 (p < 0.001). With LPS: There was a dose-dependent increase in IL-18 in the VPA groups, with significant increases in Val-75, Val-100, and Val-200 (p < 0.05). A significant difference was observed between no LPS and with LPS in all Val groups (Table 8).

Table 8 IL-18 levels at Day 3.

Without LPS	Ctrl	Phy	Val-50	Val-75	Val-100	Val-200	
Mean ± SD	163.576 ± 25.13	169.563 ± 19.46	163.055 ± 26.87	48.106 ± 13.67	80.667 ± 16.61	108.222 ± 20.93	
P-value	0.09b
0.95c
0.002*d
0.01*e
0.04*f	0.09a
0.09c
0.003*d
0.02*e
0.04*f	0.09a
0.09b
0.00*3d
0.02*e
0.04*f	0.002*a
0.003*b
0.003*c
0.45e
0.18f	0.015*a
0.018*b
0.018*c
0.45d
0.55f	0.07a
0.06b
0.04*c
0.18d
0.55e	
With LPS	Ctrl+ LPS	Phy+ LPS	Val-50+LPS	Val-75+LPS	Val-100+LPS	Val-200+LPS	
Mean ± SD	139.489 ± 23.88	122.626 ± 23.69	128.670 ± 7.82	153.195 ± 18.73	179.191 ± 18.37	182.374 ± 22.30	
P-value	0.62b-
0.68c-
0.84d-
0.34e-
0.37f-	0.62a-
0.92c-
0.48d-
0.15e-
0.17f-	0.68a-
0.92b-
0.55d-
0.18e-
0.20f-	0.84a-
0.48b-
0.55c-
0.45e-
0.48f-	0.34a-
0.15b-
0.18c-
0.45d-
0.96f-	0.37a-
0.17b-
0.20c-
0.48d-
0.96e-	
P-value for pairwise comparisons	0.670	0.401	0.441	0.004*	0.004*	0.020*	
Notes.

a P-value in comparison to Ctrl.

b P-value in comparison to Phy.

c P-value in comparison to Val-50.

d P-value in comparison to Val-75.

e P-value in comparison to Val-100.

f P-value in comparison to Val-200.

a- P-value in comparison to Ctrl+ LPS.

b- P-value in comparison to Phy+ LPS.

c- P-value in comparison to Val-50+ LPS.

d- P-value in comparison to Val-75+ LPS.

e- P-value in comparison to Val-100+ LPS.

f- P-value in comparison to Val-200+ LPS.

* Statistically significant at p ≤ 0.05.

IgA levels

Without LPS: Phy had the highest IgA levels (1.347 ± 0.42 ng/ml), significantly higher than Val-75 (p = 0.043). With LPS: IgA levels increased in all groups, but no significant differences were observed between LPS and non-LPS conditions (Table 9).

Table 9 IgA levels at Day 3.

Without LPS	Ctrl	Phy	Val-50	Val-75	Val-100	Val-200	
Mean ± SD	0.904 ± 0.860	1.347 ± 0.846	0.957 ± 0.719	0.362 ± 0.267	0.734 ± 0.531	1.119 ± 0.632	
P-value	0.43b
0.93c
0.26d
0.63e
0.98f	0.43a
0.48c
0.04*d
0.21e
0.44f	0.93a
0.48b
0.23d
0.57e
0.94f	0.26a
0.04*b
0.23c
0.51e
0.26f	0.63a
0.21b
0.57c
0.52d
0.62f	0.98a
0.44b
0.94c
0.26d
0.62e	
With LPS	Ctrl+ LPS	Phy+ LPS	Val-50+LPS	Val-75+LPS	Val-100+LPS	Val-200+LPS	
Mean ± SD	1.292 ± 1.015	1.650 ± 0.907	0.632 ± 0.571	0.899 ± 0.762	1.246 ± 1.062	1.345 ± 0.867	
P-value	0.82b-
0.23c-
0.46d-
0.90e-
0.91f-	0.82a-
0.16c-
0.35d-
0.73e-
0.74f-	0.23a-
0.16b-
0.63d-
0.28e-
0.28f-	0.46a-
0.35b-
0.63c-
0.54e-
0.53f-	0.90a-
0.73b-
0.28c-
0.54d-
0.99f-	0.91a-
0.74b-
0.28c-
0.54d-
0.99e-	
P-value for pairwise comparisons	0.565	0.998	0.473	0.335	0.357	0.662	
Notes.

a P-value in comparison to Ctrl.

b P-value in comparison to Phy.

c P-value in comparison to Val-50.

d P-value in comparison to Val-75.

e P-value in comparison to Val-100.

f P-value in comparison to Val-200.

a- P-value in comparison to Ctrl+ LPS.

b- P-value in comparison to Phy+ LPS.

c- P-value in comparison to Val-50+ LPS.

d- P-value in comparison to Val-75+ LPS.

e- P-value in comparison to Val-100+ LPS.

f- P-value in comparison to Val-200+ LPS.

* Statistically significant at p ≤ 0.05.

Discussion

In this study, we investigated the cytotoxic and inflammatory responses to phenytoin and valproic acid (VPA) on peripheral blood mononuclear cells (PBMCs), with and without bacterial lipopolysaccharide (LPS) stimulation. Our findings demonstrate that both drugs significantly affected cell viability, with VPA exhibiting a dose-dependent immunomodulatory effect. Notably, Val-75 (VPA at 75 µg/mL) showed lower cytotoxicity and reduced inflammatory responses compared to phenytoin and higher VPA doses. Additionally, all VPA treatments led to a decrease in IL-6 levels.

In contrast, phenytoin (Phy) induced higher cytotoxicity, particularly in the presence of LPS, which was reflected by significant increases in LDH levels, indicating greater cell membrane damage. Phenytoin also resulted in a marked elevation of IL-6 levels, suggesting a more pronounced pro-inflammatory effect compared to VPA. The addition of LPS exacerbated the inflammatory response, providing insights into the potential clinical implications of these drugs in chronic inflammatory conditions, such as gingival overgrowth.

We employed a combination of acute-phase (IL-6, IL-1β) and chronic-phase (IL-18, IgA) inflammatory mediators to explore the immunomodulatory effects of AEDs on human mononuclear cells under in-vitro inflammatory conditions. LPS stimulation was used to simulate inflammatory environments. We hypothesized that assessments on Day 1 would reveal immediate cellular responses, while Day 3 would capture sustained inflammatory responses induced by phenytoin and different concentrations of valproic acid. This approach aimed to model chronic inflammation, which may contribute to gingival fibrosis and overgrowth.

Chronic inflammation is a key etiological factor in gingival enlargement, where prolonged cytokine release can exacerbate tissue pathology. Therefore, in the context of our in vitro model, the analysis of cytokines on Day 3 provided insights into the ability of AEDs to either aggravate or mitigate inflammation. This may help explain the variable clinical effects of the investigated drugs on gingival overgrowth, as previously documented (Gallo et al., 2021; Lin, Guilhoto & Yacubian, 2007; Feijó Miguelis et al., 2019).

Our study found that phenytoin significantly reduced PBMC viability compared to VPA on both Day 1 and Day 3. This is consistent with the findings of Śladowska et al. (2017), who reported apoptosis and necrosis induced by phenytoin derivatives. Similar apoptotic changes in brain cells treated with phenytoin have also been observed in animal studies (Ohmori et al., 1999; Kaushal et al., 2016). Significant differences in cell count and viability were noted in control groups, both with and without LPS, suggesting that the addition of LPS to the control and phenytoin groups altered LDH activity, reflecting potential cellular responses to endotoxin exposure.

A dose-dependent cytotoxic effect of VPA was observed, with the highest concentration (Val-200) demonstrating greater cytotoxicity than the lower concentrations (Val-50 and Val-75). This aligns with previous reports indicating that serum VPA levels exceeding 100 µg/mL often lead to hematological toxicity, including leukopenia and neutropenia (Vasudev et al., 2010). High LDH levels observed in the Val-100 and Val-200 groups, both with and without LPS, further support these findings, suggesting increased cellular stress or damage at higher VPA concentrations. At lower concentrations, no significant cytotoxic effects were observed compared to controls.

The discrepancy observed between Val-100 and Val-200 on day 3 may reflect differential rates of cell detachment or early cytotoxicity at Val-200, resulting in reduced LDH levels at the time of measurement. Biological variability, including patient-to-patient differences, likely contributes to minor fluctuations in cellular responses. The difference in LDH levels between Val-100 and Val-200 is small and falls within the expected range of biological assays, suggesting it is not clinically significant but rather indicative of subtle concentration-dependent variations.

Threshold or time-dependent effects may also explain this outcome, where Val-100 exhibited more pronounced cytotoxicity on day 3, while Val-200 may have induced earlier or more complete cell death. Despite this, the overall trend aligns across assays, with LPS treatment consistently associated with lower viability and higher LDH, indicative of increased cytotoxicity, regardless of the experimental group.

The study also confirmed the dose-dependent cytotoxicity of VPA compared to phenytoin, consistent with previous ex vivo studies on myeloid progenitor cells, which demonstrated that VPA modulates neutrophil development in a concentration-dependent manner. Concentrations exceeding 100 µg/mL inhibited myeloid progenitor cell differentiation (Bartels et al., 2010; Dilek et al., 2019). PBMC viability was significantly reduced at the 200 µg/mL concentration of VPA (Val-200), which is double the upper therapeutic serum level.

Increased cytotoxic effects were observed following LPS stimulation in cells exposed to phenytoin and VPA, aligning with previous studies showing that VPA and phenytoin exhibit minimal cytotoxicity in healthy cells (Chateauvieux et al., 2010).

Our results demonstrated significantly increased IL-6 levels in the phenytoin group, both with and without LPS, compared to the control group. A similar trend was observed in cells exposed to the highest concentration of VPA (Val-200), consistent with the findings of Williamson et al. (1994), who reported elevated IL-6 levels in tissues from patients with gingival overgrowth compared to normal gingival tissues.

While previous studies (Takahashi et al., 1994; Yakovlev et al., 1996) indicated that IL-6 levels did not increase in PBMCs exposed to LPS alone, our study demonstrated that antiepileptic drugs (AEDs) stimulate PBMCs to produce IL-6. This suggests that AEDs may influence not only local gingival fibroblasts but also peripheral blood mononuclear cells, contributing to systemic inflammatory responses.

IL-6 levels were lower in cells exposed to LPS compared to non-exposed cells. This discrepancy may be attributed to IL-6’s dual role as both an anti-inflammatory and pro-inflammatory agent, depending on the physiological context (Vezzani et al., 2002). VPA, even at the highest concentrations, resulted in the lowest IL-6 production by PBMCs, both with and without LPS. The inconsistent data regarding VPA’s effects on the immune system suggest that VPA modulates immune responses differently under in vitro conditions.

The elevation in IL-6 levels with VPA was dose-dependent, with the greatest stimulation observed at 200 µg/mL of VPA (Val-200). However, LPS-stimulated groups treated with VPA at concentrations of 50, 75, and 100 µg/mL exhibited significantly reduced IL-6 levels compared to their non-LPS counterparts. These results align with the findings of Ichiyama et al. (2000), who reported that VPA significantly reduced IL-6 and TNF-α secretion in monocytes stimulated with LPS, whereas phenytoin did not. This may be attributed to VPA’s anti-inflammatory properties, which modulate cytokine concentrations by inhibiting NF-κB. Discrepancies between studies may stem from differences in experimental conditions, such as in vitro versus clinical settings, and variations in VPA dosage.

IL-1β levels were significantly elevated in the VPA groups with LPS compared to the non-LPS groups, corroborating previous research indicating that IL-1β is increased in inflamed gingival tissues (Dongari-Bagtzoglou & Ebersole, 1998). Verrotti et al. (2001) similarly reported elevated IL-1α and IL-1β levels in patients receiving anticonvulsant therapy, suggesting a role in chronic inflammation and disease progression through interactions between innate and adaptive immunity (Soria-Castro et al., 2019).

IgA levels were reduced by certain AEDs, with phenytoin notably associated with IgA deficiency (Beghi & Shorvon, 2011). While limited data exist regarding the effects of VPA on serum immunoglobulin levels, one study observed reduced IgA levels in epileptic patients undergoing VPA monotherapy, though other studies found no significant changes (Hemingway et al., 1999; Godhwani & Bahna, 2016). In our study, Val-50 significantly inhibited IgA secretion, but higher VPA concentrations and phenytoin did not produce notable alterations in IgA levels. Variations in cell type, baseline IgA levels, asymptomatic infections, sample size, population characteristics, and medication duration may account for these inconsistencies. The trend toward increased IgA levels in the phenytoin group may be linked to elevated IL-6 levels, as IL-6 plays a critical role in IgA production (Beagley et al., 1989).

Without LPS, lower induction of IL-1β was observed, particularly in the Val-75 group, suggesting a dose-specific anti-inflammatory effect under non-inflammatory conditions. This contrasts with previous studies that reported no effect of short-term VPA on inflammatory immune markers, likely due to the use of lower VPA doses (Maes et al., 1995b; Maes et al., 1995a). IL-18 levels were highest in the phenytoin (Phy) group without LPS, while VPA-treated groups exhibited lower IL-18 levels (0.19–0.39 pg/mL), supporting the potential anti-inflammatory properties of VPA. In the presence of LPS, higher VPA concentrations led to a twofold increase in IL-18 levels, consistent with in vivo studies showing elevated IL-18 following LPS injection (Ashrafi et al., 2010).

This study highlights that VPA, traditionally used for neurological disorders, may possess significant immunomodulatory properties, particularly at 75 µg/mL, by modulating pro-inflammatory and immune markers. These findings suggest potential new therapeutic applications for VPA in chronic inflammatory conditions. The dose-dependent cytotoxicity of VPA, amplified by LPS, provides insights into its broader pharmacological effects, potentially contributing to drug-induced gingival overgrowth during long-term AED treatment. Understanding VPA’s influence on cellular and molecular pathways links its role in epilepsy management to possible side effects, supporting the development of strategies to leverage its immunoregulatory benefits while minimizing adverse effects during long-term therapy.

Clinical relevance lies in the dose-specific effects of valproic acid (VPA), particularly at the Val-75 dose, which demonstrated a more favorable immunomodulatory profile compared to higher doses. Given the chronic use of VPA in epilepsy treatment, understanding its impact on gingival overgrowth and inflammatory responses is essential for improving patient care. Future research should focus on exploring the long-term effects of different VPA doses on immune modulation and soft tissue health, with the goal of identifying optimal therapeutic strategies that minimize adverse side effects, particularly in patients requiring prolonged AED therapy.

One limitation of this study is the use of a single, relatively late time point (Day 3) for cytokine analysis. While Day 3 was chosen to assess sustained inflammatory responses, it may not capture the peak secretion of some cytokines, such as IL-6. Future studies should include additional time points to capture the full dynamics of cytokine release across both acute and chronic phases of inflammation. Larger sample sizes are also needed to more comprehensively assess the influence of factors such as age, sex, genetic background, and lifestyle on the observed effects. The exclusion of gingival fibroblast cells, which play a crucial role in tissue homeostasis, may have limited our understanding of the drugs’ impact on extracellular matrix production and remodeling. Additionally, the in vitro conditions used in this study may not fully replicate the complexity of the in vivo environment (Shi et al., 2019). Finally, further research at the gene level could provide deeper insights into the mechanisms behind changes in LDH activity and the observed immunomodulatory effects.

Conclusion

This study demonstrates distinct immunomodulatory and cytotoxic effects of phenytoin (Phy) and valproic acid (VPA) on human monocytic cells. Phy significantly decreased cell viability and increased cytotoxicity, while VPA exhibited a dose-dependent effect, with higher concentrations leading to increased cytotoxicity. Phy elevated IL-6 levels, indicating a pro-inflammatory effect, whereas VPA generally suppressed IL-6 production. VPA at 75 µg/ml reduced LPS-induced cytotoxicity and inflammation, lowering LDH release and pro-inflammatory cytokines (IL-1β, IL-6, IL-18) as well as IgA. These findings suggest that VPA at lower concentrations may provide a therapeutic window for modulating inflammation in relevant disease contexts. Further research is needed to explore VPA’s immunomodulatory mechanisms and its clinical potential in chronic inflammatory conditions, bacterial infections, and gingival enlargement.

Supplemental Information

Supplemental Information 1 Raw data

Additional Information and Declarations

Competing Interests

Author Contributions

Human Ethics

Data Availability

The authors declare there are no competing interests.

Aminah Alesawy conceived and designed the experiments, performed the experiments, analyzed the data, prepared figures and/or tables, authored or reviewed drafts of the article, and approved the final draft.

Norah Alotaibi performed the experiments, authored or reviewed drafts of the article, and approved the final draft.

Marwa Alalshaikh performed the experiments, authored or reviewed drafts of the article, and approved the final draft.

Faisal E. Aljofi performed the experiments, prepared figures and/or tables, authored or reviewed drafts of the article, and approved the final draft.

Nada Aldossary conceived and designed the experiments, performed the experiments, authored or reviewed drafts of the article, and approved the final draft.

Nada Al-Zahrani performed the experiments, authored or reviewed drafts of the article, and approved the final draft.

Omar Omar conceived and designed the experiments, performed the experiments, analyzed the data, prepared figures and/or tables, authored or reviewed drafts of the article, and approved the final draft.

Marwa Madi conceived and designed the experiments, performed the experiments, analyzed the data, prepared figures and/or tables, authored or reviewed drafts of the article, and approved the final draft.

The following information was supplied relating to ethical approvals (i.e., approving body and any reference numbers):

Institutional Review Board of Imam Abdulrahman Bin Faisal university (IRB-PGS-2023-02-467)

The following information was supplied regarding data availability:

The raw measurements are available in the Supplementary File.

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
