# Peer review of "Impact of phenytoin and valproic acid on cytotoxicity and inflammatory mediators in human mononuclear cells: with and without lipopolysaccharide stimulation"

_PeerJ, doi:10.7717/peerj.19102_

## Round 0.1 · original submission · Major Revisions

The manuscript, in its current form, is poorly written and requires major revisions. Please address the corrections suggested by the reviewers. Additionally, it is crucial to provide details regarding the sample size for each group and the number of times the experiments were repeated.

What was the rationale behind choosing the specific time points for analysis? For example, it is known that IL-6 peaks 6 hours after stimulation and subsequently decreases.

All chemicals were diluted in DMSO; did the control cells receive the vehicle (DMSO) as well?

The results section should be rewritten to avoid unnecessary repetition of data or numerical values from the graphs.

The conclusion is missing info about phenytoin findings.

Finally, the raw data file should include the raw data itself, and not the calculated mean values.

·

Basic reporting

Review report on “Cytotoxicity and inflammatory response of valproic acid and phenytoin on human mononuclear cells: an in-vitro study”
This study “Cytotoxicity and inflammatory response of valproic acid and phenytoin on human mononuclear cells: an in-vitro study” aimed to investigate the effect of phenytoin and valproic acid on PBMC viability and expressions of inflammatory mediators in the presence and absence of bacterial lipopolysaccharides. This manuscript is poorly written, and the authors have just carried out a few experiments which are insufficient for this manuscript to be accepted for publication.
My remarks related to this manuscript are:
1. The authors should provide supplementary detail with proper references about cytotoxicity and inflammatory response in the introduction section.
2. The last paragraph of the introduction should highlight the main objectives of the current study.
3. The author should cite (Beghi & Shorvon, 2011) at once, I did not understand why the authors have cited this reference at the end of every sentence again and again.
4. Method section is poorly written, rewrite it with proper methodology and supported by previously reported related studies.
5. How was the sample size defined? Ideally, a priori sample size calculation should be performed to determine the appropriate sample size. Include a statement in the methods section.
6. Reference for ...……Peripheral blood mononuclear cells isolation …... in the material and methods section should be provided.
7. Include catalog numbers for all resources in the methods section.
8. How have the authors calculated the concentration of phenytoin and valproic acid in the material and method section line # 151-155? Please clarify how the concentration has been calculated.
9. Proper method should be provided with references for …….Total, live, and dead cell count, viability assessment and ELISA……...in the method section.
10. The authors have not rearranged the references throughout the manuscript according to the journal guidelines.
11. Include a statement regarding the blinding of the experimenter or the lack thereof in the methods section.
12. In statistical analysis, authors said that all statistical tests are done by SPSS and Graph Pad prism. Please specify the statistical results appropriately in all results clearly mentioning the parameters used.
13. Results should be strongly discussed with proper statistics and p-value.
14. The titles of the results and figure legends should state the main findings of the study directly.
15. Explain each figure properly and write up the figure legend with proper statistics and p value.
16. Result section is too confusing, the authors should use simple and clear sentences to understand the main finding of this study.
17. Please write the proper units related to the previous research papers of IL-6, IL-1β, IL-18 and IgA on y-axis of graphs.
18. The results of the tests for normality and variance homogeneity should be reported for each experiment.
19. For bar graph presentations, please show exact data points (e.g. dot plots) rather than bar plots throughout the manuscript.
20. There are several grammatical and typographical mistakes throughout this paper. Please consult a native speaker to improve its clarity.
21. The discussion should start with a short paragraph summarizing the main findings of the study.
22. The authors should follow the same size and style of font throughout the manuscript.
23. I did not understand why the authors have conducted this study, what is the novelty and future perspective of this study.
24. The authors should clearly discuss the conclusion of this study. The conclusion is too confusing.
25. Whole manuscript did not justify the title of the current study.
26. Layout diagram of experiment should be suitable for readers, clearly providing the summary of the whole manuscript as the authors mentioned only grouping in layout.

Experimental design

Poor

Validity of the findings

The whole manuscript needs substantial revisions.

Reviewer 2 ·

Basic reporting

The paper reports the cytotoxicity of valproic acid (4 concentrations) and phenytoin (only therapeutic concentration) on PBMC cells from 4 donor blood samples. The paper appears to be logically organized, with a clear abstract, well laid-out introduction, a graphical presentation of the results, and a discussion that compares the results with the literature. English is also acceptable.
In the methodology section, when explaining the statistical analysis, explain which experimental values were compared against which, to obtain the p values reported in the text and the figures. For example, did you compare all values against the control, or did you compare val100 against phenytoin as well?
The results section can be improved by reporting the results in tables. Put the results of LDH levels, standard deviations, and p-values in a table. The text is too hard to follow. Include the units. Similarly, results of cell counts, cell viability, IL-6, and other experimental results should be tabulated. Do not repeat the values in the text, unless there is an important issue about the values that you want to discuss.
Authors should comment on the observed higher toxicity of Val-100 than Val-200 on day 3 of the LDH test with LPS, or repeat the test since this result does not align with the cell concentration results.

Experimental design

Blood from donors has been mixed. It would have been interesting to identify interindividual variability.

Validity of the findings

The findings are valid. Novelty is average.

---

## Round 0.2 · accepted · Accept

The authors made the necessary corrections, significantly improving the quality of the manuscript.

Reviewer 2 ·

Basic reporting

This new version is much better in terms of the scientific writing. The text is clear and much easier to read.

Experimental design

The manuscript reports research with well-defined questions and appropriate methodology that, in the new version, have been reported in an acceptable format.

Validity of the findings

Conclusions are based on the statistically tested observations.